# A mosaic monoploid reference sequence for the highly complex genome of sugarcane

Olivier Garsmeur [1,2], Gaetan Droc [1,2], Rudie Antonise[3], Jane Grimwood[4], Bernard Potier [5], Karen Aitken[6], Jerry Jenkins[4], Guillaume Martin [1,2], Carine Charron [1,2], Catherine Hervouet[1,2], Laurent Costet [7], Nabila Yahiaoui[1,2], Adam Healey[4], David Sims[4], Yesesri Cherukuri[4], Avinash Sreedasyam[4], Andrzej Kilian[8], Agnes Chan[9], Marie-Anne Van Sluys [10], Kankshita Swaminathan[4], Christopher Town [9], Hélène Bergès[11], Blake Simmons [12], Jean Christophe Glaszmann[1,2], Edwin van der Vossen[3], Robert Henry [13], Jeremy Schmutz[4,14] & Angélique D'Hont[1,2]

Sugarcane (*Saccharum* spp.) is a major crop for sugar and bioenergy production. Its highly polyploid, aneuploid, heterozygous, and interspecific genome poses major challenges for producing a reference sequence. We exploited colinearity with sorghum to produce a BAC-based monoploid genome sequence of sugarcane. A minimum tiling path of 4660 sugarcane BAC that best covers the gene-rich part of the sorghum genome was selected based on whole-genome profiling, sequenced, and assembled in a 382-Mb single tiling path of a high-quality sequence. A total of 25,316 protein-coding gene models are predicted, 17% of which display no colinearity with their sorghum orthologs. We show that the two species, *S. officinarum* and *S. spontaneum*, involved in modern cultivars differ by their transposable elements and by a few large chromosomal rearrangements, explaining their distinct genome size and distinct basic chromosome numbers while also suggesting that polyploidization arose in both lineages after their divergence.

[1] CIRAD (Centre de Coopération Internationale en Recherche Agronomique pour le Développement), UMR AGAP, F-34398 Montpellier, France. [2] AGAP, Univ Montpellier, CIRAD, INRA, Montpellier SupAgro, 34060 Montpellier, France. [3] KEYGENE N.V., 6708 PW Wageningen, The Netherlands. [4] HudsonAlpha Institute for Biotechnology, Huntsville, AL 35801, USA. [5] SASRI (South African Sugarcane Research Institute), Mount Edgecombe 4300, South Africa. [6] CSIRO (Commonwealth Scientific and Industrial Research Organisation), St. Lucia, QLD 4067, Australia. [7] CIRAD, UMR PVBMT, F-97410 Saint-Pierre, La Réunion, France. [8] Diversity Arrays Technology, Yarralumla, ACT 2600, Australia. [9] J. Craig Venter Institute, Rockville, MD 20850, USA. [10] Universidade de Sao Paulo, Sao Paulo 05508-090 SP, Brazil. [11] INRA-CNRGV, 31326 Toulouse, Castanet-Tolosan, France. [12] JBEI Joint BioEnergy Institute, Emeryville, CA 94608, USA. [13] QAAFI (Queensland Alliance for Agriculture and Food Innovation), University of Queensland, St. Lucia, QLD 4072, Australia. [14] Department of Energy, Joint Genome Institute, Walnut Creek, CA 94598, USA. Correspondence and requests for materials should be addressed to A.D'H. (email: dhont@cirad.fr)

Sugarcane—a member of the Poaceae family—produces 80% of the world's sugar and has recently become a primary crop for biofuel production[1]. Cultivars are vegetatively propagated, while sugarcane breeding is still essentially focused on conventional methods since sugarcane genetics knowledge has lagged behind that of other major crops[2]. This is due to the interspecific, polyploid, and aneuploid nature of modern sugarcane cultivar genomes, a complexity that exceeds that of most if not all other crops. Modern sugarcane cultivars are derived from a few interspecific hybridizations performed a century ago between *S. officinarum* and *S. spontaneum*, two highly polyploid species. *S. officinarum* ($2n = 8× = 80$, $x = 10$)[3,4] has a high sugar content and is believed to have been domesticated around 8000 years ago from the wild species *S. robustum* (mainly $2n = 60$, $80$, and up to $200$)[5,6]. *S. spontaneum* is a wild species with various cytotypes and many aneuploid forms ($2n = 5× = 40$ to $16× = 128$; $x = 8$)[3,7] and was used to incorporate disease resistance, vigor, and adaptability[4]. Recovery of high sugar production was achieved by backcrossing the first hybrid to *S. officinarum* and then enhanced via $2n$ chromosome transmission by the female *S. officinarum*[8,9]. The resulting cultivars are highly heterozygous, aneuploid, and have 100–130 chromosomes, most of which are derived from *S. officinarum*, 10–20% from *S. spontaneum*, and ~10% from interspecific recombinants[10,11] (Fig. 1).

In the last 20 years, molecular geneticists have achieved significant advances in developing molecular resources and enhancing the overall understanding of the sugarcane genome. Several geneticmaps have been produced (reviewed in ref. [12]) mainly based on single-dose markers that are the most informative markers in this high polyploidy context[13]. None of these maps were saturated, but they revealed that chromosome assortment in modern cultivars results from general polysomy with some cases of preferential pairing[14–21], while the meiosis of modern sugarcane cultivars mainly involves bivalent pairing[8,22,23]. Meanwhile,

comparative mapping with other Poaceae species revealed extensive genome-wide colinearity with sorghum, which thus became a model for sugarcane[15,24–27]. QTL studies have been performed to identify genomic regions involved in important agronomic traits, generally revealing minor effects[28–31], except for the presence of a few major resistance genes[20,32–34]. New approaches are now being developed, such as genome-wide association (GWA) studies[35–37] and genomic selection[38], both of which exploit the high level of linkage disequilibrium observed within sugarcane cultivars[39,40]. Genomic resources such as large expressed sequence tag (EST) libraries[41] and BAC libraries[42,43] have been developed. A sugarcane reference genome would greatly help to achieve further progress on molecular genetics and functional genomics geared toward assisting in the development of improved cultivars. However, sequencing such a highly complex genome poses challenges that have not been addressed in any prior sequencing project due to the high polyploidy, aneuploidy, and interspecific structure of this genome, with a complete set of hom(oe)ologs predicted to range from 10 to 12 copies[44,45]. This polyploidy results in a total genome size of about 10 Gb for sugarcane cultivars, while the monoploid genome size is about 800–900 Mb, close to that of sorghum (750 Mb)[46].

In this context, an attractive initial strategy would be to assemble the gene-rich, recombinationally active part (i.e., euchromatin) of a monoploid (i.e., basic) sugarcane genome. Sorghum and rice genome sequencing showed that euchromatin is gene-rich, repeat-poor, and accounts for most of recombination, while being of similar physical size in sorghum and rice (250–300 Mb) despite their overall genome-size difference (450 Mb for rice and 750 Mb for sorghum)[47]. Comparative analysis among sugarcane hom(oe)ologous chromosome segments (BAC) showed very high gene colinearity and sequence conservation[48,49]. One haplotype sequence would thus provide a good reference for the other haplotypes (=hom(oe)ologous chromosomes). In addition, these studies revealed a high level of microcolinearity between sorghum and sugarcane[47–51]. Sorghum could be used as a guide to identify a minimum tiling path (MTP) of BACs covering the euchromatin of a monoploid genome of sugarcane.

While new sequencing and assembling methods tailored for complex genomes continue to be developed, we deployed this original strategy to produce a first sugarcane reference genome sequence assembly.

This international initiative focused on the R570 cultivar, which has the best-characterized sugarcane genome to date[10,17,18,21,24,44,45]. This cultivar was obtained by CERF (now eRcane) in Réunion (http://www.ercane.re/). It has broad adaptability and is used as a parent stock in many breeding stations worldwide, while also being commercially very successful in Réunion, Mauritius, and Guadeloupe. The genome of this cultivar has about 115 chromosomes, including 10% of whole chromosomes derived from *S. spontaneum* and 10% from *S. officinarum*/ *S. spontaneum* recombinant chromosomes, with the remaining being whole chromosomes from *S. officinarum*[10]. Genetic maps of this cultivar were constructed[17,18,21] (http://tropgenedb.cirad. fr/tropgene/JSP/index.jsp) and aligned with sorghum[24]. A BAC library of 103,296 clones[43] representing 14× monoploid genome coverage and 1.3× total genome coverage of this cultivar[44] is available and has been used by the research community for comparative structural analyses[26,47–49,52,53].

Here, we implemented this strategy using WGP[TM] technology[54] to align sugarcane BACs from the R570 cultivar with the sorghum genome and identify a MTP of 4660 sugarcane BACs originating from distinct hom(oe)ologous chromosomes that we sequenced, assembled, and annotated. We developed a SNP-based genetic map for the R570 cultivar and performed global

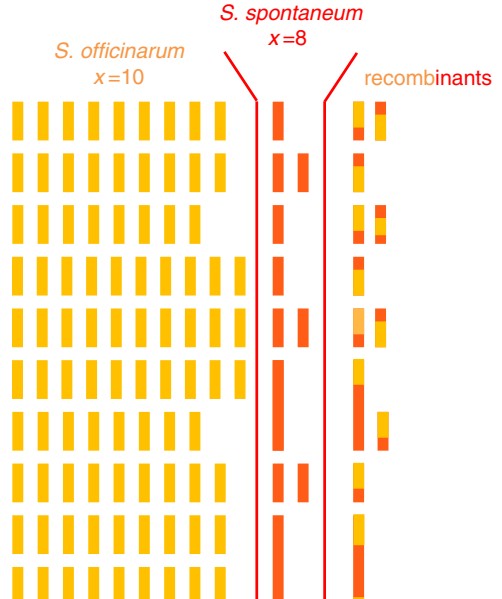

**Fig. 1** Schematic representation of the genome of a typical modern sugarcane cultivar. Each bar represents a chromosome, in orange or red when originating from *S. officinarum* or *S. spontaneum*, respectively. Chromosomes aligned on the same row are hom(oe)ologous chromosomes (HG). The key characteristics of this genome are the high polyploidy, aneuploidy, bispecific origin of the chromosomes, the existence of structural differences between chromosomes of the two origins, and the presence of interspecific chromosome recombinants

*S. officinarum*
*x* =10

*S. spontaneum*
*x* =8

recombinants

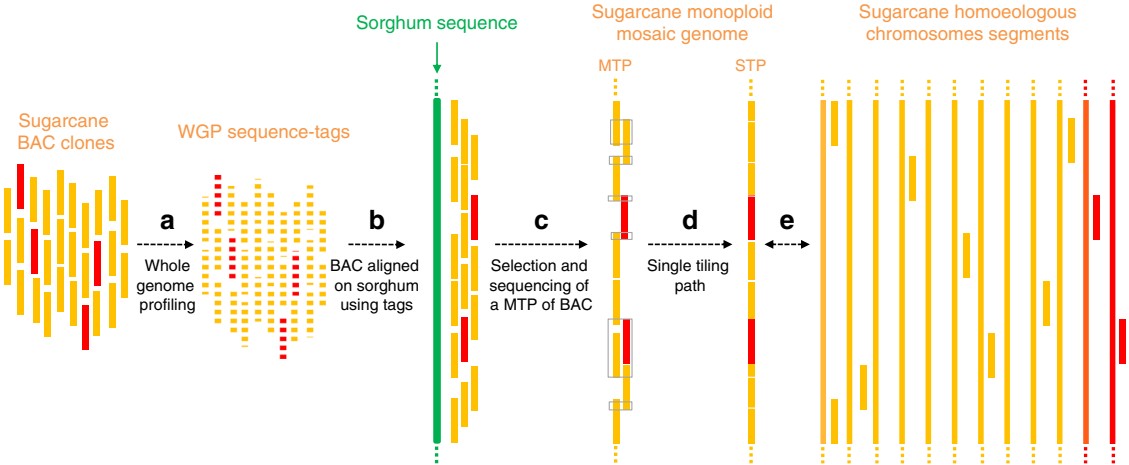

**Fig. 2** Sequencing strategy targeting the sugarcane monoploid genome based on the overall synteny and colinearity conservation within sugarcane hom (oe)ologs and with sorghum. **a** WGP sequence tags were produced from R570 sugarcane BACs. **b** WGP sequence tags were aligned onto the sorghum sequence, thus allowing the location of sugarcane BACs on sorghum. **c** A minimum tiling path of a BAC (MTP) corresponding to a monoploid sugarcane genome was defined and sequenced. **d** Overlapping BAC sequences were trimmed to construct the single tiling path (STP). **e** The STP sequence contains BAC contigs that belong to distinct hom(oe)ologous chromosomes. *S. officinarum* and *S. spontaneum* chromosome segments are represented in orange and red, respectively

| Table 1 Selection and sequencing of BACs targeting the gene-rich part of the sugarcane monoploid genome | | | | | | |
|---|---|---|---|---|---|---|
| **Selection and sequencing of a sugarcane minimum tiling path (MTP) of BACs** | | | **Sugarcane single tiling path (STP)** | | | |
| **Sorghum chromosome (Mb)** | **Nb of sugarcane BACs anchored** | **Nb of BACs sequenced (Mb)** | **Mosaic super scaffolds (Mb)** | **Genes** | | **TE** |
| | | | | **Nb** | **%** | **%** |
| Sb01 (81) | 1924 | 778 (94) | Sh01 (67) | 4614 | 15 | 44 |
| Sb02 (78) | 1598 | 594 (68) | Sh02 (49) | 3270 | 13 | 43 |
| Sb03 (74) | 1624 | 634 (74) | Sh03 (51) | 3540 | 14 | 43 |
| Sb04 (69) | 1261 | 496 (56) | Sh04 (42) | 2881 | 14 | 44 |
| Sb05 (72) | 827 | 289 (33) | Sh05 (22) | 1337 | 11 | 41 |
| Sb06 (61) | 1060 | 404 (48) | Sh06 (33) | 2189 | 13 | 45 |
| Sb07 (66) | 871 | 305 (36) | Sh07 (28) | 1903 | 13 | 42 |
| Sb08 (63) | 623 | 265 (31) | Sh08 (23) | 1381 | 12 | 43 |
| Sb09 (59) | 891 | 391 (46) | Sh09 (34) | 2143 | 13 | 44 |
| Sb10 (61) | 1053 | 379 (45) | Sh10 (32) | 2058 | 16 | 43 |
| 683 Mb | 11,732 | 4535 (531 Mb) | 382 Mb | 25,316 | 13 | 43 |

genome-wise comparisons between *S. officinarum*, *S. spontaneum*, and sorghum. This first sugarcane sequence represents the gene-rich part of the monoploid sugarcane genome.

## Results

**A MTP for the gene-rich part of the genome**. The overall strategy that we used to produce a sugarcane monoploid reference sequence assembly is represented in Fig. 2. A total of 20,736 BAC clones from the sugarcane R570 cultivar, representing around two-fold the monoploid genome of sugarcane, were analyzed via WGP. This technology generates Illumina short-read sequences (WGP tags) from BAC clone-restriction fragments. A total of 701,066 WGP tags were obtained and those that were common to several BACs were discarded. Half (222,745) of the remaining WGP tags could be aligned onto the sorghum sequence. Among them, 90,953 WGP tags aligning at a single position allowed anchoring of 11,732 sugarcane BACs onto the ten sorghum chromosomes (Table 1). Another 577 already-sequenced BAC clones were also anchored onto the sorghum genome. The sugarcane BAC clones were, not surprisingly, mostly distributed in sorghum gene-rich distal chromosomal regions (Fig. 3) as mainly genes are conserved between the two species. Among a

total of 12,309 anchored BACs, an MTP of 4660 BAC clones, corresponding to the minimal set of BACs providing the best coverage of the gene-rich part of the 10 sorghum chromosomes, was defined (Table 1 and Supplementary Table 1).

**A 382-Mb mosaic sequence with 25,316 predicted gene models**. The 4083 BAC clones of the MTP for which no sequences were available were sequenced in pools of 24 or 96 using PacBio RSII technology with a sequencing depth of around 100× per BAC (Supplementary Table 1). BAC pools were designed to avoid pooling hom(oe)ologous overlapping BACs that could complicate the assembly of individual BACs. A total of 5896 PacBio contigs were assembled with a mean length of 102,858 bp. Contigs were assigned to each of the BAC clones using their corresponding WGP tags. The vast majority of BAC clones were assembled in one contig (85%) or a few contigs (Supplementary Table 1). A total of 4535 BAC sequences were obtained, including the 577 BAC clones that were already sequenced, corresponding to 531 Mb of sequence (Table 1 and Supplementary Table 1). Figure 4 illustrates the extensive coverage of the sorghum genome by the sugarcane BAC sequences. The MTP contained sequence overlaps between adjacent BAC clones that were then trimmed to

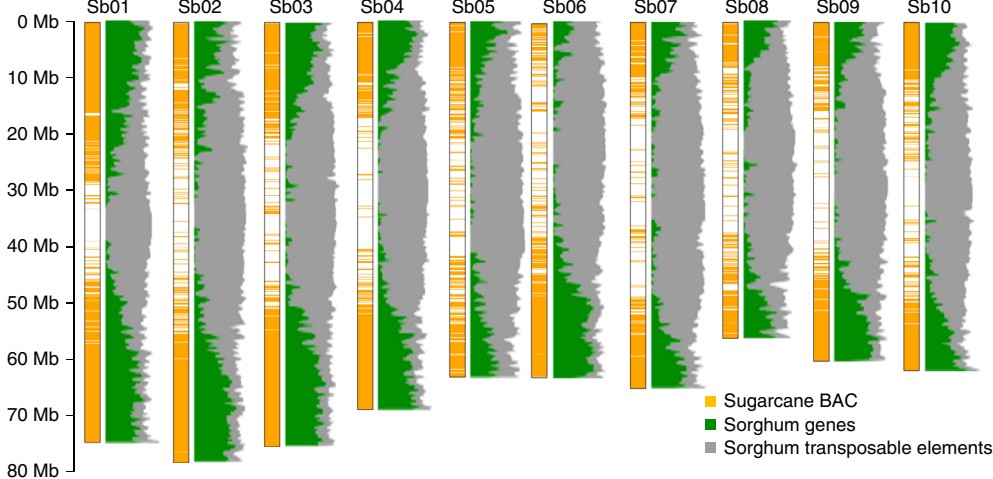

**Fig. 3** Distribution of the 11,732 sugarcane BACs aligned onto the sorghum genome through WGP. Sugarcane BAC clones are represented by orange bars. Sorghum gene and transposable element densities are represented in green and gray, respectively

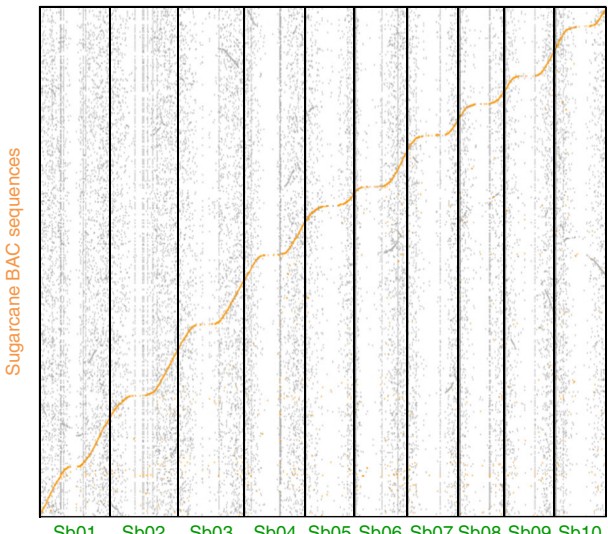

**Fig. 4** Coverage of the sorghum genome by the sequenced sugarcane BAC MTP. Dot plot with alignment of BAC sequences from the sugarcane MTP (y-axis) on the sorghum genomic sequence (x-axis). The sorghum genome covered by the sugarcane MTP sequence is highlighted in orange. Gray segments correspond to duplicated regions resulting from ancestral whole-genome duplications in Poales

construct a single tiling path (STP) that is a single copy in the sugarcane gene space. Based on sequence comparisons with WGS data from *S. officinarum* and *S. spontaneum*, we could assign an *S. officinarum* origin to 71% of the STP vs. 23% to *S. spontaneum*. These proportions were consistent with the overall estimation of *S. officinarum* vs. *S. spontaneum* chromosome constitution (~85 vs. ~15%) of cultivar R570 based on molecular cytogenetics[10]. BAC contigs from this STP were oriented and merged into 10 mosaic super-scaffolds based on synteny conservation with sorghum. This STP consisted of 382 Mb of high-quality sequence in 3965 contigs, with an average size of 96 kb, and including only 0.02% uncalled bases (N).

A total of 25,316 protein-coding gene models were predicted, which represented 13% of the STP sequence (Table 1, Supplementary Table 2). A sorghum orthologous gene could be found for 20,809 (82%) of these sugarcane-predicted genes among the

27,532 sorghum protein-coding gene models predicted on the sorghum genome. Many of the remaining 4507 gene models were short monoexonic predictions with no hits in the Uniprot/Trembl sequence database (http://www.uniprot.org) and no Interpro domains and likely corresponded to overpredictions. Among the 1267 gene models that had at least one Interpro domain, the most frequently found domains corresponded to proteins involved in signaling or defense responses such as kinases, receptor-like kinases, and NB-ARC proteins. Proteins involved in protein degradation by the proteasome were also found, specifically F-box proteins and proteins with an SKP1/BTB/POZ domain (Supplementary Table 3). Many of these proteins are encoded by genes that are known to evolve rapidly[55,56].

Among a set of 9871 conserved grass genes, 79% (7791) were captured by the STP.

Gene colinearity conservation at the BAC level (i.e., ~100 kb) was analyzed by comparing BACs of the sugarcane STP to the corresponding orthologous regions in sorghum. Among the 20,809 sugarcane-predicted gene models for which an ortholog could be found in the sorghum sequence, 83% were located within the corresponding orthologous sorghum segments, thus leaving 17% of sugarcane-predicted genes in the non-collinear position in sorghum. Since we selected the BACs to be sequenced based on their global alignments with sorghum, this level of non-colinearity could be higher in more rearranged regions between sugarcane and sorghum, or among sugarcane hom(oe)ologs.

Finally, based on a sequence comparison of 13,069 pairs of orthologous genes, we evaluated the sugarcane/sorghum divergence at 8.5 My (median ks = ~0.111), which is in the upper range of previous estimations[48–51].

**Differential transposable element (TE) amplification among *Saccharum* species.** TEs were annotated in the STP and their proportions estimated in WGS data from the R570 cultivar, one *S. spontaneum* accession and one *S. officinarum* accession (Table 2). They represented 43% of the STP as compared to at least 52% for the whole R570 cultivar genome in agreement with the STP focusing on the gene-rich euchromatin fraction of the genome as opposed to the TE-rich heterochromatin. The most represented TE families in both *S. officinarum* and *S. spontaneum* are the LTR retrotransposon families Gypsy Chromovirus and Copia Maximus/SIRE. The insertion dates of the complete LTR retrotransposons displayed an L shape, revealing continuous transposition activity over the last few million years

**Table 2 Transposable element (TE) contents in the single tiling path (STP) and in whole-genome sequencing data from the R570 cultivar, one *S. officinarum* (S. OFF) and one *S. spontaneum* (S. SPONT)**

| | R570 STP annotation | | | Repeat explorer clusters assembled from WGS data | | | |
|---|---|---|---|---|---|---|---|
| | STP BAC | S. OFF BAC | S. SPONT BAC | STP reads | R570 WGS | S. OFF WGS | S. SPONT WGS |
| TE (all) | 43.1% | 47.3% | 38.3% | 41.2% | 50.9% | 53.6% | 46.5% |
| DNA transposons TIR | 4.6% | 4.9% | 4.3% | 1.8% | 1.8% | 1.9% | 1.9% |
| *DTA_hAT* | 2.79% | 2.96% | 2.68% | 0.24% | 0.24% | 0.23% | 0.28% |
| *DTC_CACTA* | 1.15% | 1.23% | 1.02% | 0.71% | 0.82% | 0.86% | 0.81% |
| *DTM_Mutator* | 0.51% | 0.53% | 0.51% | 0.65% | 0.43% | 0.43% | 0.53% |
| *DTX* | 0.18% | 0.21% | 0.12% | 0.25% | 0.34% | 0.37% | 0.28% |
| LTR retrotransposons | 36.8% | 40.9% | 32.3% | 29.8% | 38.1% | 40.1% | 34.6% |
| Copia | 16.09% | 17.53% | 14.44% | 13.34% | 14.01% | 15.25% | 12.09% |
| *RLC_Alel_Retrofit* | 2.51% | 2.42% | 3.01% | 1.43% | 0.66% | 0.57% | 0.83% |
| *RLC_Angela* | 1.09% | 1.19% | 1.00% | 1.01% | 0.95% | 0.99% | 0.80% |
| *RLC_Bianca* | 0.02% | 0.02% | 0.03% | 0.15% | 0.13% | 0.13% | 0.15% |
| *RLC_Ivana_Oryco* | 0.16% | 0.16% | 0.14% | 0.37% | 0.22% | 0.25% | 0.15% |
| *RLC_Maximus/SIRE* | 9.53% | 10.74% | 7.81% | 10.02% | 11.69% | 12.92% | 9.82% |
| *RLC_TAR* | 0.11% | 0.11% | 0.14% | 0.25% | 0.24% | 0.25% | 0.31% |
| *RLC_Tork* | 0.04% | 0.06% | 0.00% | 0.11% | 0.11% | 0.13% | 0.03% |
| *RLC* | 2.63% | 2.83% | 2.31% | 0.00% | 0.00% | 0.00% | 0.00% |
| Gypsy | 20.39% | 23.06% | 17.51% | 16.46% | 24.10% | 24.80% | 22.54% |
| *RLG_Athila* | 1.42% | 1.32% | 1.99% | 1.71% | 3.47% | 3.27% | 5.23% |
| *RLG_Chromovirus* | 11.00% | 13.29% | 7.38% | 10.47% | 16.65% | 17.62% | 11.32% |
| *RLG_Ogre_TAT* | 4.84% | 4.85% | 5.65% | 4.27% | 3.98% | 3.91% | 5.98% |
| *RLG_Reina* | 0.84% | 0.95% | 0.69% | 0.00% | 0.00% | 0.00% | 0.00% |
| *RLG* | 2.29% | 2.64% | 1.80% | 0.00% | 0.00% | 0.00% | 0.00% |
| RLX | 0.35% | 0.35% | 0.39% | 0.00% | 0.00% | 0.00% | 0.00% |
| Unclassified | 1.6% | 1.4% | 1.7% | 9.6% | 10.9% | 11.6% | 10.0% |

(Supplementary Fig. 1). TE represented a larger part of the *S. officinarum* genome compared to the *S. spontaneum* genome (~54 vs. ~47%), in agreement with their distinct basic genome sizes of ~950 vs. ~ 800 Mbp[46,57]. The larger basic genome of *S. officinarum* mainly resulted from a more intense amplification of the Gypsy Chromovirus and Copia Maximus/SIRE families in this species' lineage after its divergence from *S. spontaneum* around 1.5–3.5 Mya[48,53]. The majority of TE families are less abundant in *S. spontaneum* compared to *S. officinarum*, with the exception of two Gypsy families, Ogre_TAT and Athila that have been amplified more intensively in *S. spontaneum* after the divergence of these two species.

**Chromosome structural variations among *Saccharum* species.** A SNP-based genetic map was built using 186 individuals from a self-progeny of cultivar R570 genotyped by DArTseq. A total of 12,468 single-dose SNPs were assembled into 132 cosegregation groups (CGs) that included 5–624 DArTseq markers, with an average of 94 markers.

The origin of CGs was then tentatively assigned to ancestral species (*S. officinarum* vs. *S. spontaneum*) based on 5377 SNPs differentiating *S. officinarum*/*S. robustum* vs. *S. spontaneum* accessions within a panel of 34 accessions genotyped with DArTseq. Most CGs (74%) were assigned to *S. officinarum* and 26% were assigned to *S. spontaneum* or to recombinant chromosomes between these two species (Fig. 5). CGs assigned to *S. officinarum* were generally much smaller (i.e., contained fewer markers) than CGs assigned to *S. spontaneum*, as also observed in previous sugarcane-mapping studies[18,58]. This is explained by the fact that the *S. spontaneum* genome is less redundant than the *S. officinarum* genome in sugarcane cultivars and thus bears more single-dose markers amenable to genetic mapping.

Synteny conservation between sugarcane CGs and sorghum chromosomes was analyzed using 5406 DArTseq markers from the R570 genetic map that could be aligned with the sorghum sequence. Most sugarcane CGs (125, 95%) had a large majority of markers aligned with one single sorghum chromosome (Fig. 5a, b, Supplementary Fig.2), while for seven CGs a large majority of markers aligned with two distinct sorghum chromosomes (Fig. 5c). Sugarcane CGs were assembled in hom(oe)ology groups (HGs, one HG = one basic chromosome set) based on these syntenic relationships (Fig. 5 and Supplementary Figs. 2, 3).

Interestingly, all seven CGs that aligned with two sorghum chromosomes were assigned to *S. spontaneum* chromosomes or to recombinant chromosomes between *S. spontaneum* and *S. officinarum* (Fig. 5). These results suggested that a few large chromosome rearrangements differentiate *S. spontaneum* on one side and *S. officinarum* and sorghum on the other. In *S. spontaneum*, one HG was orthologous to sorghum chromosome 2 and one arm of sorghum chromosome 8, whereas another HG was orthologous to sorghum chromosome 9 and the other arm of sorghum chromosome 8. A third *S. spontaneum* HG was orthologous to sorghum chromosome 6 and one arm of sorghum chromosome 5. The R570 SNP map did not feature any *S. spontaneum* CG aligned with the second arm of sorghum chromosome 5. However, preliminary SNP mapping data on another sugarcane clone (MQ 76-53, Supplementary Fig. 3) revealed a few CGs orthologous to sorghum chromosome 7 and part of sorghum chromosome 5 (Fig. 5). In addition, Aitken et al.[15] also observed a few CGs in cultivar Q165 orthologous to chromosomes 5 and 7 of sorghum. This suggests that one *S. spontaneum* basic chromosome could be orthologous to sorghum chromosome 7 and one arm of sorghum chromosome 5. This configuration may have been lacking in the R570 genetic map, probably because our map is far from saturation, and CGs only partially cover the corresponding chromosomes. In addition, one

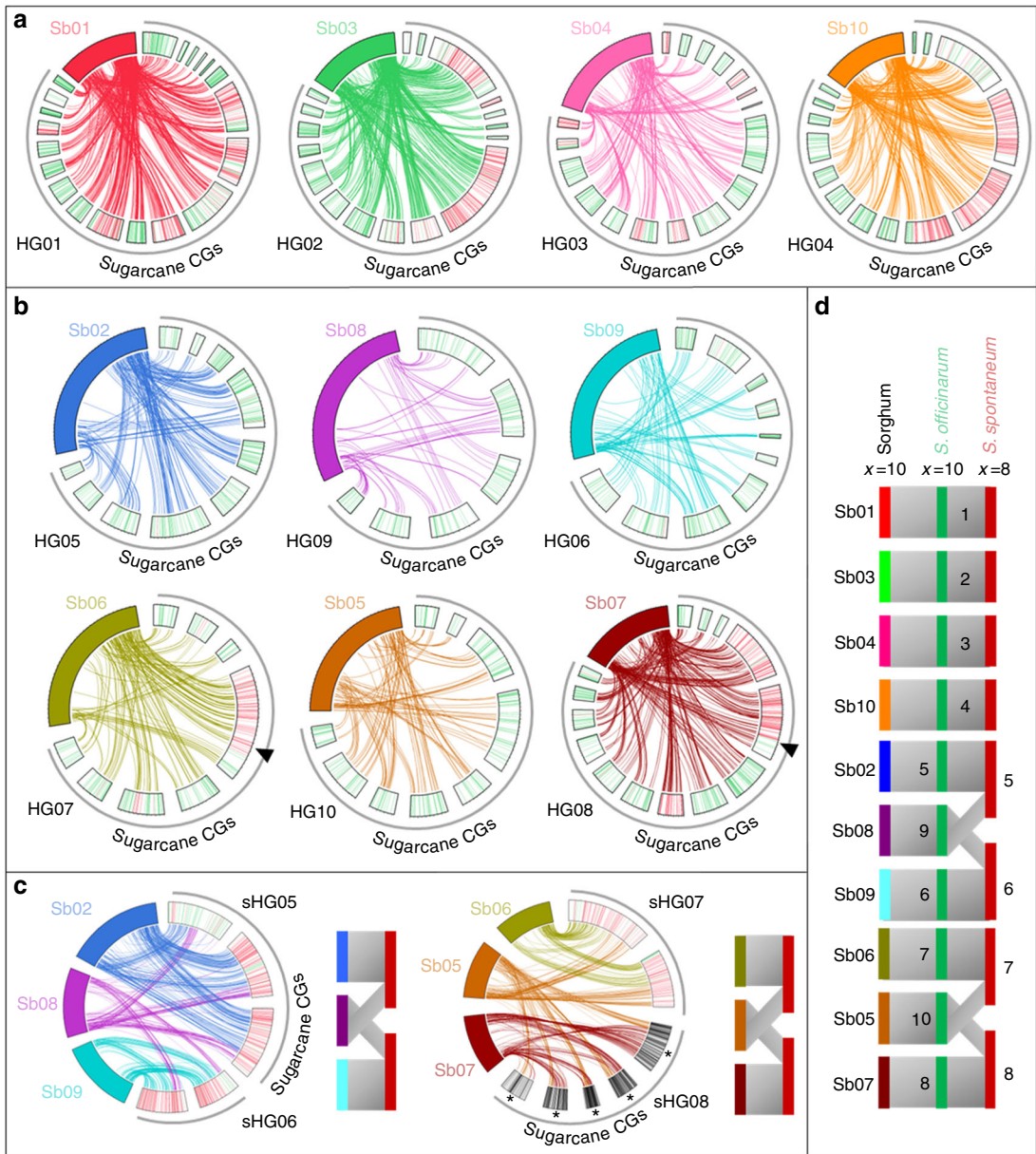

**Fig. 5** SNP-based sugarcane genetic map with putative origin of cosegregation groups and comparison with sorghum chromosomes. The 132 CGs of cultivar R570 are represented with SNP markers assigned to *S. officinarum* or *S. spontaneum* indicated by green and red bars, respectively. Circos represents orthologous relationships between sugarcane CGs and sorghum chromosomes (Sb1–Sb10) based on the alignment, for each CG, of a majority of the markers on one (**a**, **b**) or two (**c**) sorghum chromosomes (color links) (see Supplementary Fig. 2 for representation of all links). Based on these orthologous relationships, CGs were assembled in hom(oe)ology groups (HGs): **a** Four HGs (HG1–HG4) including CGs from *S. officinarum*, *S. spontaneum*, and interspecific recombinants orthologous to one sorghum chromosome. **b** Six HGs (HG6–HG10) including *S. officinarum* CGs, and a few *S. spontaneum* and interspecific recombinant CGs orthologous to one sorghum chromosome. Arrows point to two interspecific recombinations (see text). **c** Two pairs of HGs (sHG5 and sHG6, and sHG7 and sHG8), each including *S. spontaneum* or interspecific recombinant CGs orthologous to two sorghum chromosomes. **d** Schematic comparison of the deduced basic genome organization in *S. spontaneum* ($x = 8$), *S. officinarum* ($x = 10$), and sorghum ($x = 10$). Asterisk represents CGs from the MQ76-53 SNP map (see text and Supplementary Fig. 3)

large CG assigned to HG08 in cultivar R570 apparently resulted from an interspecific recombination event between *S. spontaneum* and *S. officinarum* chromosomes: this recombination could have removed the portion orthologous to sorghum chromosome 7 from the original *S. spontaneum* chromosome (see the arrow in Fig. 5b). A similar situation was observed for one interspecific recombinant CG assigned to HG07 (see arrow in Fig. 5b).

Previous mapping studies have suggested a few large chromosome structural variations between sugarcane and sorghum with some sugarcane CGs having segments orthologous to two sorghum chromosomes[15,24,59]. While our genetic map remains unsaturated, our relatively high number of markers with assignments to *S. officinarum* vs. *S. spontaneum* enabled us for the first time to describe chromosome rearrangements involving two sets of three ancestral chromosomes that were each rearranged in two chromosomes (Fig. 5). These rearrangements corroborated and explained a variation in basic chromosome numbers in *S. officinarum* vs. *S. spontaneum*, i.e., $x = 10$ and $x = 8$, respectively[3,60]. The coexistence of distinct chromosome organizations in the genome of modern cultivars most likely has an

impact on their meiosis and may be a source of the aneuploidy observed in modern cultivars. A basic chromosome number of $x = 10$ is found in both sorghum and *S. officinarum* and is recognized as being ancestral in Saccharinae-Sorghinae[50]. These rearrangements must have arisen in the *S. spontaneum* lineage after its divergence from the lineage of *S. officinarum* and its recognized wild ancestor *S. robustum*. The mechanisms involved will require further investigation but could imply two successive reciprocal translocations, as reported in Brassicaceae and Poaceae[61,62]. This scenario also suggests that polyploidization occurred independently in *S. officinarum/S. robustum* and *S. spontaneum* lineages after their divergence around 1.5 to 3.5 Mya[48,53] in discordance with previous hypotheses[50].

## Discussion

The sequence reported here is the first assembly of the sugarcane genome with 382 Mb of sequence in 3965 high-quality contigs. This was achieved by isolating BAC clones corresponding to a monoploid genome of sugarcane to overcome assembly difficulties linked to the high ploidy, aneuploidy, and heterozygosity of sugarcane cultivars. This sequence assembly is proposed as a reference for the euchromatin gene-rich part of the genome, known to account for most of the recombination, and thus representing the most useful part for breeding.

We confirmed the overall gene synteny conservation and colinearity between sugarcane and sorghum, while identifying 17% of genes predicted to be non-collinear in the surveyed regions. We showed that the two species involved in modern cultivars differ for LTR retrotransposon amplification after their divergence, and that the larger basic genome size of *S. officinarum* compared to *S. spontaneum* is mainly due to the amplification of two families, i.e., the Gypsy RLG_Chromovirus and Copia RLC_Maximus/SIRE. In addition, we showed that these two species differed by two rearrangements, each involving three sets of ancestral chromosomes, which explains their distinct basic chromosome numbers. Two chromosome organizations thus coexist in the genome of modern cultivars. We propose a chronologic sequence for these rearrangements and polyploidization in the *Saccharum* genus.

This monoploid reference sequence provides an essential genome template for aligning sequencing data, such as genotyping by sequencing, WGS, and RNA-Seq data to explore hom(oe)ologous allelic variation and perform genetic (e.g., QTL and GWAS) and genomic studies in cultivars and sugarcane germplasm. These studies, as before, will require methods (e.g., SNP calling) adapted to the high polyploid context of sugarcane. This reference will also facilitate the identification of candidate genes/loci in regions found to be associated with targeted phenotypic traits by permitting the location of these regions within the annotated genome. Finally, it will provide an essential framework to help current whole-genome sequencing initiatives meet the challenge of assembling such a complex genome.

## Methods

**Sugarcane BAC selection through whole-genome profiling**. WGP of 20,736 BACs from the R570 sugarcane BAC library[43] was performed as described by Van Oeveren et al[54]. A total of 701,066 WGP tags of 50 bp, corresponding to 18,843 BACs, were obtained, representing an average of 37 WGP tags per BAC. WGP tags common to distinct BACs were discarded and the remaining 455,656 WGP tags were aligned with the sorghum genome using Bowtie2 v.2.3.2 (http://bowtie-bio.sourceforge.net/bowtie2/index.shtml) with the following parameters: -D 15, -R 2, -N 1, -L 22, and -i S, 1 and 0.75. Around 50% of the tags (232,911) aligned with the sorghum sequence (V2.1). A total of 90,953 WGP tags (20%) aligned at a single position and 141,958 WGP tags (30%) aligned in multiple positions. Sugarcane BACs were first anchored onto the sorghum genome using tags that aligned at single positions. A BAC was anchored if at least 20% of the tags (representing a minimum of three tags) obtained from this BAC could be mapped in a 300-kb window. A total of 11,732 BACs could be anchored onto the sorghum genome

(Supplementary Table 1). Then 22,299 WGP tags that aligned at multiple positions on sorghum but in the same 300-kb window as the anchored BACs were used as additional information to refine BAC positions. Another 577 already-sequenced BAC clones were anchored onto the sorghum genome using BLAST[63]. Finally, among the 12,309 anchored BACs, an MTP of 4660 BAC clones was selected based on visual inspection of their position on the sorghum genome and of BAC clone overlaps.

**BAC sequencing and assembly**. An initial set of 577 BAC clones were sequenced individually, including 245 BAC clones reported by De Setta et al.[52], Garsmeur et al.[49], and Jannoo et al.[48], and 332 BAC clones sequenced by DOE-JGI using $2 \times 250$-bp paired-end single-indexed Illumina and assembled with Phrap (Green, http://www.phrap.org/phredphrap/phrap.html). For 4083 BAC clones, DNAs were extracted individually and then pools of BAC DNA were obtained. PACBIO RS II was used for sequencing with a targeted 100× depth per BAC. Half of the BACs were sequenced by KeyGene in pools of 24 clones and the other half by DOE-JGI in pools of 96 clones. Assembly was performed using HGAP3 (version 2.3.0) followed by consensus-sequence calling with Quiver (version 2.1). For each sequenced pool, WGP tags corresponding to the pooled BACs were aligned with the contigs obtained. BLAST alignments of WGP tags with identity and coverage equal to or greater than 99% were used to assign contigs to the corresponding BAC clones, resulting in the assembly of 3958 BAC sequences. Whole BAC sequence contigs were aligned to the sorghum sequence using the BLAST algorithm with the best hits being retained to produce Fig. 4.

To construct STP, contigs (larger than or equal to 20 kb) from the 4535 sequenced BAC clones were aligned with the sorghum-annotated nucleotide gene set (version 3.1) using BLAT (https://github.com/dib-lab/ged-docs/wiki/BLAT, parameters -noHead -extendThroughN -q=rna -t=dna). Sorghum gene alignments were used to inform the process of reducing the initial MTP in an STP representing a single copy of sugarcane gene space. BAC segments sharing at least two sorghum genes were considered as redundant overlapping regions. These BAC clones were aligned with one another using Gepard (http://cube.univie.ac.at/gepard). Overlapping BAC regions sharing identical sorghum genes were then collapsed into a single path, thus maximizing the number of retained genes and minimizing sequence loss. Sorghum gene positions were finally used to order, orient, and join the BAC contigs using 10,000 Ns to form 10 super-scaffolds.

**Ancestral origin of sugarcane BAC clones**. To identify the origin, *S. officinarum* vs. *S. spontaneum*, of sugarcane BAC sequences, Illumina WGS data from one *S. officinarum* accession (LA Purple) and one *S. spontaneum* accession (SES234B) were aligned separately to BAC sequences using BWA-MEM. The cumulative read coverage of each BAC clone was calculated and compared using SAMtools (min depth = 5 reads, min quality = 20) and bedtools genomecov (-bga). BAC contigs were then classified if there was at least 10% difference in the total coverage of the BAC clone by *S. officinarum* or *S. spontaneum* reads. Among the 4535 BAC contigs sequenced, 86% could be assigned with 78% to *S. officinarum* and 22% to *S. spontaneum*. The remaining BAC contigs (>50 kb) were analyzed using species-specific 24-bp kmers (24 mers) extracted from (i) repetitive regions (read depth > $2 \times SD$ of mean), and (ii) well-covered regions (read depth within 1 SD of the mean) of the previous set of *S. officinarum* and *S. spontaneum* BAC contigs assigned to *S. officinarum* and *S. spontaneum*. These kmers were used to mask BAC contig sequences, and the ancestral origin was determined based on the majority of sequences masked by the species-specific kmers. Finally, a total of 4281 BAC contigs were classified, representing 360 Mbp (94.4%) out of the 382 Mbp of the STP, with 272 Mb (71%) assigned to *S. officinarum* and 88 Mb (23%) to *S. spontaneum*.

**R570 RNA-Seq data**. RNA was extracted from R570 cultivar leaves, roots, and stems using a method adapted from Bugos et al[64]. Three cDNA libraries (one per tissue) were built using the TruSeqStranded mRNA sample preparation kit (Illumina) and subsequently paired-end sequenced ($2 \times 125$ bp) on an Illumina HiSeq 2500 system. A total of 354 million reads were produced and 63, 54, and 57 million paired-read clusters were built for leaves, roots, and stems, respectively, and assembled using Trinity in 340,409 transcripts larger than 300 bp (https://github.com/trinityrnaseq/trinityrnaseq/wiki).

**Sugarcane genes annotation**. Structural annotation was performed using EuGene 4.2a (http://eugene.toulouse.inra.fr), which combines evidence from alternative splice site detection, sequence model training, genome masking, transcript mapping, and protein homology to define confident gene models. The R570 cultivar RNA transcripts, together with transcripts from six sugarcane genotypes produced by Cardoso-Silva et al.[65], were aligned on the sugarcane STP using GMAP (https://github.com/juliangehring/GMAP-GSNAP/) and best-scoring hits were kept. The resulting alignments were integrated as sugarcane transcript evidence in the prediction process. Protein sequences from the reference sorghum proteome (Phytozome, release 3.1.1) were aligned with the sugarcane STP using BLASTX to obtain protein homology evidence.

Gene function annotation was assigned based on sequence and domain conservation. Protein sequences were aligned with SwissProt, TrEMBL, and

sorghum datasets by BLASTP using an E-value threshold of 1e-10. Best-hit BLAST results were then used to define the gene functions. Moreover, InterProScan = (https://www.ebi.ac.uk/interpro/search/sequence-search) was performed to annotate protein domains, extending the annotation to Gene Ontology terms associated with these protein domains. The predicted gene models were finally screened to remove models that could correspond to TEs using CENSOR (https://www.girinst.org/censor/)[66] resulting in a total of 25,316 sugarcane genes annotated on the STP.

**TEs annotation**. Sets of 2 million Illumina reads (150 bp) were retrieved from WGS data for the R570 cultivar, one *S. officinarum* accession (LA Purple) and one *S. spontaneum* accession (SES 234). A similar dataset was created from the sugarcane STP through generation of in silico Illumina reads with WGSIM (https://github.com/lh3/wgsim). These four datasets were used to perform a graph-based clustering analysis of repetitive sequences using the RepeatExplorer pipeline[67]. A total of 179 clusters were obtained, manually inspected using the .htlm summary output file generated by the pipeline, and clusters corresponding to chloroplast or containing less than 3000 Illumina reads were not considered for further analyses. The 135 remaining clusters were manually annotated using a library of 58 previously annotated sugarcane elements[52] and using similarity searches against Repbase (https://www.girinst.org/repbase/), a protein domain database of plant mobile elements (included in RepeatExplorer). For each annotated cluster, the WGS read numbers and percentage were calculated in datasets corresponding to STP, R570, *S. officinarum*, and *S. spontaneum* (Table 2).

TE annotation of the sugarcane STP was performed with the TEannot pipeline from the REPET package v2.5 (https://urgi.versailles.inra.fr/Tools/REPET), which requires a reference library of TEs. This latter reference library was built through clustering of the following TE sequences:

—The 58 previously annotated sugarcane TEs.
—1883 full-length LTR retrotransposons detected in the sugarcane STP using LTR_FINDER v1.05 (https://github.com/xzhub/LTR_Finder).
—The 135 RepeatExplorer clusters described above.

A total of 446 clusters were obtained using UCLUST v1.2.22q with the following parameters: --id 0.80 --maxlen 25000 --minlen 100 –nucleo (https://drive5.com/usearch/manual/uclust_algo.html) and used for TE annotation with the REPET package.

The percentage of TE in the STP was calculated based on the sequence proportion covered by each TE type (Table 2).

The timing of insertion of full-length LTR retrotransposons was determined by estimating the amount of divergence between LTR sequences. The two LTRs of each retrotransposon were aligned using MAFFT (https://mafft.cbrc.jp/alignment/software/), and the genetic divergence was estimated using the Kimura 2 parameter method. The insertion dates were estimated using a substitution rate of 1.3e-08 per site per year[68].

**DArTseq genotyping**. A mapping population encompassing 186 individuals from a self-progeny of cultivar R570 and a panel of 34 accessions belonging to the *Saccharum* genus were genotyped through DArTseq at Diversity Arrays Technology Pty Ltd., Australia. Two adapters corresponding to two different restriction enzyme overhangs were used; a PstI-compatible adapter, including a barcode region for multiplexing and a reverse NspI-compatible adapter. Only fragments that contained both adapters (PstI-NspI) were used to build sequencing libraries (Illumina, USA). The resulting libraries (two copies per sample) were sequenced in batches of 94 per lane on Illumina Hiseq2500.

The raw sequencing data obtained were then demultiplexed using GBSX (https://github.com/GenomicsCoreLeuven/GBSX), filtered for quality using FASTQC (https://www.bioinformatics.babraham.ac.uk/projects/fastqc/), and barcode adapters were removed using Cutadapt (http://opensource.scilifelab.se/projects/cutadapt/). Only sequence reads ≥ 30 bp were kept for further analysis, representing 885 million reads for the mapping population and 161 million reads for the diversity data, corresponding to an average of around 5 million reads per sample.

**Construction of the R570 cultivar SNP-based genetic map**. Identification of high-confidence single-dose SNP markers: DArTseq data from the 186 individuals from the self-progeny of R570 were analyzed to identify high-confidence single-dose SNP markers useful for genetic mapping. The analysis was based on a de novo approach involving the assembly of a pseudomolecule that war further used as a template for read mapping. This pseudomolecule was assembled by clustering all DArTseq reads using CD-Hit-EST[69] with the following parameters: -c 0.9, -al 0.9, -p 1, and -M 0. Clusters were then randomly assembled to form the pseudomolecule, with each cluster being separated by a stretch of 1000 "N". A pipeline (available at https://github.com/SouthGreenPlatform/VcfHunter/) was developed to perform the following complete process: (1) DArTseq reads were aligned on the pseudomolecule using BWA-MEM (http://bio-bwa.sourceforge.net); (2) a local realignment of reads around indels was performed with the GATK IndelRealigner (https://software.broadinstitute.org/gatk/documentation/tooldocs/3.8-0/org_broadinstitute_gatk_tools_walkers_indels_IndelRealigner.php); (3) metrics were generated at each single-nucleotide position within realignments (count number of A, C, G, and T at each position) with Bam-readcount (https://github.

com/genome/bam-readcount); (4) SNP positions on the pseudomolecule and allele count were converted in a standard variant-calling format file (VCF); and (5) the VCF was then screened to determine the genotype (homozygous vs. heterozygous) of each individual and at each position. Several criteria were taken into account to improve the genotype determination accuracy. First, a 30× minimum and 1000× maximum read depth was required and only diallelic polymorphic markers were considered. For heterozygous genotypes, the minor allele then had to be represented by at least two reads with a frequency of at least 4%. Ambiguous genotypes with a minor allele frequency ranging from 1 to 4% were converted as missing data (uncalled genotype). Homozygous genotypes were determined when no variant was observed or with a frequency of <1% (considered as sequencing error). (6) The segregation ratio of single-dose polymorphic markers within the population was then evaluated using a $\chi^2$ test ($p = 0.05$). SNP markers displaying an expected 3:1 ratio and with less than 20% missing data were considered as high-confidence single-dose SNPs. A total of 13,062 single-dose SNPs were identified.

Construction of the R570 cultivar SNP-based genetic map and alignment with the sorghum genome: The genetic map was built with the 13,062 single-dose SNP markers using JoinMap 4.1[70] with a LOD score threshold of 10 and a maximum recombinant fraction of 0.35. CGs containing at least five markers were kept. Markers were ordered with MSTmap[71] with standard parameters.

DArTseq markers from sugarcane CGs were aligned with the sorghum genome using BLASTN and 5406 DArTSeq markers with hit identity and coverage equal to or greater than 95% were kept. Orthologous relationships between sugarcane CGs and sorghum chromosomes were then established based on the proportion of DArTseq markers from one sugarcane CG aligned with each of the sorghum chromosomes. A minimum of five markers, representing at least 10% of aligned markers for a CG was used. For 125 CGs, a large majority of markers (mean = 88%) aligned with a single sorghum chromosome. For seven CGs, a large majority of markers (mean = 81%) aligned with two sorghum chromosomes. The resulting genetic map with markers aligned with the sorghum chromosomes was constructed using Circos[72] (Fig. 5 and Supplementary Figs. 2, 3).

**S. officinarum vs. S. spontaneum origin of R570 cultivar CGs**. DArTseq data from the panel of 34 *Saccharum* accessions, comprising 12 *S. officinarum* accessions (Ashy Mauritius, Badila, Banjarmasin Hitam, Black Cheribon, BNS-3066, Crystallina, EK28, IJ76-432, IS76-203, LA Purple, Lousier, and NG57-123), nine *S. robustum* (IJ76-445, IM76-234, IS76-138, Mol-4503, NG28-251, NG77-021, NG77-054, NG77-108, and NG77-230), and 13 *S. spontaneum* accessions (Glagah-1286, Glagah-WT, IK76-041, IK76-067, Mandalay, Mol-5904, NG51-2, Coimbatore, SES-014, SES-178, SES-208, SES-264, and SES-289A), were aligned on the same pseudomolecule developed for the analysis of DArTseq genetic mapping data. A raw VCF file comprising the SNP positions and allele counts was created and filtered, as described above for identifying high-confidence SNP markers (pipeline, steps 1–5). A SNP was considered as originating from *S. officinarum* if present in at least two *S. officinarum* or *S. robustum* accessions and absent from all *S. spontaneum* accessions. Reciprocally, a SNP that was present in at least two *S. spontaneum* accessions and absent from all *officinarum* and *robustum* accessions was considered as originating from *S. spontaneum*. This enabled the determination of the ancestral origin of 5377 SNP markers from the R570 genetic map.

**Sugarcane–sorghum comparative analysis**. Identification of orthologous gene pairs: Sorghum gene models from the v2.1 DOE-JGI annotation set (http://phytozome.jgi.doe.gov/) were filtered to remove those corresponding to TE-like genes using CENSOR, as done previously for sugarcane genes. For genes with several predicted alternative transcripts, we only kept the longest one. The resulting 27,532 sorghum genes were aligned with the 25,316 annotated sugarcane genes using BLASTP, and orthologous gene pairs with identity equal to or greater than 70% and coverage equal to or greater than 40% were retained. Since both genomes were annotated with different pipelines, some genes that had been predicted in one genome could have been missed in the other genome. We checked if genes for which no orthologs were found using BLASTP could be retrieved in the genomic sequences. These genes were thus aligned with genomes using TBLASTN and the matching regions were realigned using EXONERATE[73] (model protein2genome: bestfit' and cutoff identity equal to or greater than 70%). A second complementary approach was performed using BLAT.

Set of core grass genes: Orthologous genes between *Sorghum bicolor* v3.1 and ten other grass genomes *Brachypodium distachyon* v3.1, *B. stacei* v1.1, *Oryza sativa* v7_JGI, *Oropetium thomaeum* v1.0, *Panicum hallii* var FIL2 v2.0, *P. virgatum* v4.1, *Setaria italica* v2.2, *S. viridis* v1.1, *Zea mays* Ensembl-18, and *Z. mays* PH207 v1.1 were downloaded from Phytozome (https://phytozome.jgi.doe.gov/pz/portal.html). Phytozome uses the stand-alone InParanoid 4.1 software[74], a BLAST-based algorithm to compute protein homology analysis with use_bootstrap set to 1 and the default parameters for other options to identify orthologous genes between each genome pair. *S. bicolor* genes with orthologs in all of the aforementioned grass species were considered as core grass genes in this study. Primary protein sequences (with the longest transcript) for core grass genes were obtained from Phytozome for further analysis.

Microcolinearity of sugarcane and sorghum genes: BAC sequences from the sugarcane STP were aligned with BLASTN on the sorghum genome to identify orthologous genomic regions between the two genomes. Then the gene content was

compared for each orthologous region. Sugarcane/sorghum orthologous gene pairs located in the expected orthologous regions were considered as colinear genes, while orthologous gene pairs located in regions that differed from those expected were considered as non-collinear genes.

Divergence timing: Protein sequences from sugarcane gene models were aligned with the sorghum proteome using BLASTP and best bidirectional mutual hits were retained. The 17,704 resulting orthologous gene pairs were realigned using CLUSTALW (https://www.ebi.ac.uk/Tools/msa/clustalw2/). These alignments were used to guide nucleic coding sequence alignments with PAL2NAL (http://www.bork.embl.de/pal2nal/). Synonymous substitution rates (Ks) were calculated using the Nei–Gojobori method implemented in the PAML package. All of these steps were automated with an available Python script (synonymous_calc.py, https://github.com/tanghaibao/bio-pipeline/tree/master/synonymous_calculation). Divergence time (T) was then estimated using $T = MKs/2k$, where MKs represents the median Ks (0.1268) calculated from 17,704 orthologous gene pairs and $k$ is the substitution rate of $6.5 \times 10^{-9}$ per site and per year[75].

**Code availability**. The pipeline developed to perform the complete process of SNP mapping, calling, and filtering is available at https://github.com/SouthGreenPlatform/VcfHunter/.

**Data availability**. BAC and STP sequences, as well as gene annotations are available on the sugarcane genome hub (http://sugarcane-genome.cirad.fr). BAC sequences have also been deposited in the European Nucleotide Archive (ENA) at the EMBL-European Bioinformatics Institute under accession number ERZ654945.

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

## Acknowledgements

We thank the International Consortium for Sugarcane Biotechnology for financial support of this work. This work was supported by the South Green Bioinformatics platform (http://southgreen.cirad.fr). The work conducted by the Joint BioEnergy Institute and the Joint Genome Institute was supported by the Office of Science of the U. S. Department of Energy under Contract No. DE-AC02-05CH11231 with the Lawrence Berkeley National Laboratory.

## Author contributions

O.G. performed WGP analysis, MTP selection, TE annotation, genetic mapping analysis, and comparative analyses with sorghum. E.V. and R.A. produced WGP data. J.G., B.P., K.A., R.A., D.S., M.-A.V.S., H.B., B.S., E.V., R.H., and J.S. performed BAC DNA extraction, pooling, sequencing, and assembly. M.-A.V.S. provided a set of annotated TE. K.S. provided WGS data. B.P., C.C., and C.H. produced RNA-Seq data. G.D., N.Y., and O.G. performed protein-coding gene annotation and analyses. A.S. identified the set of core grass genes. G.D. performed website development. J.J., Y.C., and A.H. assembled the STP and assigned contigs to ancestral species. L.C., C.H., A.K., G.M., A.C., C.T., and O.G. performed plant material development, DNA extraction, DArTseq and GBS genotyping, and SNP calling. A.D.H. and O.G. interpreted data and wrote the paper. J.-C.G., N.Y., and J.S. revised the manuscript. A.D.H. designed and coordinated the project.

## Additional information

**Competing interests:** The authors declare no competing interests.

