## [Peer Review File · Nature Communications]

Reviewers' comments:

Reviewer #1 (Remarks to the Author):

Comments for the manuscript entitled "A monoploid reference sequence for the highly complex genome of sugarcane"

Sugarcane is the most important source of sugar production, but the complexity of its genome hinders effective research for its genetic improvement. In this manuscript, the authors aligned sugarcane BACs to sorghum and built a monoploid reference genome for sugarcane. In addition, DarTseq genotyping was performed in the progenies of R570, from which a genetic map was constructed and large scale structural variations were identified. This is the first step to a high-quality sugarcane genome. I have a few concerns as follows.

1. In general, a reference genome includes all of the sub genomes even though the species is polyploidy. In the manuscript, only a monoploid reference was constructed and this is a mosaic from all sub genomes of the 12X genome. I have not figured out how useful this is for future research, e.g. SNP calling and GWAS. For SNP calling, all polymorphism will stack onto the monoploid genome and yield lots of spurious SNP calls. I would not believe those SNPs will have trustworthy LD, which is fundamental to GWAS. Please enlighten me and readers.
2. Following the previous comments, is it possible to map sequenced BACs to the genetic map generated in the paper, so these BACs can be separated to sub genomes?
3. The placement of BACs basically relies on alignment of BAC end sequences. It can be confusing when placing BACs using sequence information only, since there are too many duplications and transposons in the genome. Genetics map is useful and complementary to the sequence-based tools. I would suggest the authors to take a look at the agreement between the genetic map and tilling path of 4535 BACs.
4. In the smaller scale, I am wondering why the authors didn't barcode those STP BACs and sequence them independently. It would achieve higher assembly accuracy by doing so, just like those old-time but highly accurate reference genomes using the BAC-by-BAC approach.
5. Two sequencing approaches were used to assembly BACs. If it is possible, I would compare the results of two approaches. For example, taking a few BACs which was sequenced by PacBio then sequence them using Illumina, then compare the results. It would be help to decide how to assemble a high-quality sugarcane genome in future, e.g. high density genetic markers + BACs sequences by Illumina or PacBio on the whole genome.
6. In line 281, it goes too far to announce that "is the first high quality assembly of sugarcane genome". In contrast, it needs to be emphasized the genome coverage is very limited and the monoploid reference genome is a mixture of all sub genomes.

Reviewer #2 (Remarks to the Author):

The authors have generated a reference resource of 4083 gene enriched BAC reference assemblies using PACBio sequencing for sugarcane. The strategy used 20,736 BAC end sequences, and aligned to sorghum to select a minimum tiling path 11,732 genes enriched BACs, plus 577 previously sequenced BACs. From these 4,660 were selected to represent a sequencing tiling path, of gene rich regions, representing one of the 8 homeologous regions. 4083 BACs were pooled, and then sequenced using PAC Bio, for a total of 531 MB of assembled contigs. Contigs were mapped back to the individual BACs using initial tag sequence. Overlap between BACs were identified, trimmed and the contigs were then merged to a pseudomolecule based on sorghum order.

The authors present a large body of work, have been careful in their analysis, and have generated a valuable resource for the community.

As a genome resource I have concern regarding the representation of the mosaic nature of the pseudomolecules generated representing a heterogeneous mixture of Homeologous groups. This is very misleading giving the polyploidy nature of the genome. Would it be better to represent these as the individual BACs? And provide the tiling path information to be used versus pseudomolecules?

While the data is available from the French site the BAC contigs and annotations should be submitted to ENA or GenBank.

The authors discuss gene space and comparison to sorghum, but do not present statistics on the gene structure stats, length of transcript, exon, and intron, UTRs.

Is there any annotation on the TES and do they see differences between the two ancestral genomes identified by the mapping population. For the two ancestral genomes, based on the contigs available, can the authors say anything about the fractionation of genes between these?

The author's state this will be a useful resource for the fine mapping, are they referring to the anchored BAC map or the sequencing itself? If the latter, it will be useful for only a small fraction of the genome, given the coverage of the orthologous group, while the BAC map can be addressed for fine mapping and walking if needed.

The authors also point to the use of analyses for expression data. Were the authors able to provide insights on how the minimum genome space impacts transcription profiling studies, using the data sets generated on the project.

Major suggestions:

- 1) Submission of BACs & annotations to ENA.
- 2) Gene statistics table comparison to sorghum. Consider using Sorghum V3 annotations recently published.
- 3) TE annotations and comparison to sub genomes
- 4)

Minor suggestions:

- 1) Figure 1 was cited in the introduction, would recommend citing this as a result.
- 2) Review language Line 108-110
Genetic maps of this cultivar were constructed, and unpublished data, and aligned with sorghum.
- 3) Table 1 may be better represented as a follow, chart with detail numbers in a supplementary table.
- 4) Assembly summary and statistics could be moved to the assembly result section to give the readers a better overview.

Reviewer #3 (Remarks to the Author):

The genome of sugarcane is difficult to assemble due to its polyploid, aneuploid and heterozygous features. To get around some of these difficulties, Garsmeur et al. have assembled a monoploid sugarcane genome using BAC sequencing and sorted the assembly into pseudomolecules based on synteny with sorghum. Construction of a genetic map using DArTseq markers showed the presence of chromosomal rearrangements. The manuscript is well-written and the level of English is high.

The methods used by the authors are sound and follow standard practice for genome assembly, though perhaps notably additional mapping technologies such as HiC and BioNano or a denser SNP-based genetic map may have allowed higher mapping resolution in complex areas.

Sugarcane is an important crop and even though this is a monoploid assembly, it may be of interest to readers of Nature Communications, though currently it reads as a rather technical description of the assembly. The manuscript could be improved with additional biological analysis. For example, other recent genome assemblies published include analyses of intraspecific gene presence-absence variation, or additional analysis of key gene functions and traits which in this case would include sugar production among others. By placing the genome assembly in a biological context while acknowledging the limitations of this not being a complete assembly would in my opinion improve the manuscript.

Minor comments:

It will be good to specify the version of sorghum genome used.

It will be good to specify the version of each software used, such as bowtie2, HGAP3, Quiver, BLAST, BLAT and Trinity.

In bowtie2 parameter settings on line 312, the '-i' option should be S, 1, 0.75 instead of S1, 0.75?
L45 Reference recommended.

L59 Parentheses around "review in 11". Should read "reviewed in"?

L66 "resistant" should be "resistance"?

L313-318 Can be confusing to read, should be reworded for clarity.

Reviewers' comments:

Reviewer #1 (*Remarks to the Author*):

Comments for the manuscript entitled "A monoploid reference sequence for the highly complex genome of sugarcane"

Sugarcane is the most important source of sugar production, but the complexity of its genome hinders effective research for its genetic improvement. In this manuscript, the authors aligned sugarcane BACs to sorghum and built a monoploid reference genome for sugarcane. In addition, DarTseq genotyping was performed in the progenies of R570, from which a genetic map was constructed and large scale structural variations were identified. This is the first step to a high-quality sugarcane genome. I have a few concerns as follows.

1. In general, a reference genome includes all of the sub genomes even though the species is polyploidy. In the manuscript, only a monoploid reference was constructed and this is a mosaic from all sub genomes of the 12X genome. I have not figured out how useful this is for future research, e.g. SNP calling and GWAS. For SNP calling, all polymorphism will stack onto the monoploid genome and yield lots of spurious SNP calls. I would not believe those SNPs will have trustworthy LD, which is fundamental to GWAS. Please enlighten me and readers.

The sugarcane cultivar genome is organized very distinctly from that of classical allopolyploids such as wheat or cotton. The sugarcane genome contains chromosomes from two species with globally a polysomic inheritance, i.e. each chromosome can pair with any other one, including pairing between chromosomes of the two species (although some cases of preferential pairing can be observed) (see Jannoo et al 2004 *Heredity* 93:460-467). In addition, the genome is aneuploid so it is expected that each class of chromosome can contain various numbers of homolog/homeolog chromosomes (see new figure 1). So it corresponds more to a *S. officinarum* polysomic polyploid genome introgressed with around 15% of chromosomes from *S. spontaneum* (another polysomic polyploid). We added a figure to better explain this (new Figure 1).

Classically, reference sequences for diploid genomes are based on a double haploid accession, or one homozygous accession, so they represent one haplotype/one monoploid genome within a species that contains many distinct haplotypes with substantial variation (as exemplified recently with the description of core and pangenomes). Then the sequences of the germplasm/GWAS population analyzed are mapped on this reference. For allopolyploid genomes such as wheat, which behaves more as a juxtaposition of three diploid genomes, the same is done with the three homeologous genome versions used as reference.

Because of the polysomic polyploidy characteristics of sugarcane genome, SNP calling is performed differently. A reference (sorghum monoploid genome before, the sugarcane monoploid reference in the future) is used to map all sequence reads and then SNP calling is performed to search for SNPs within the population/germplasm analyzed. Authors working on sugarcane have used conventional tools such as GATK, TASSEL,... (e.g. Yang et al 2017a, *Mol breeding* and 2017b *BMC Genomics*; Balsalobre et al 2017 *BMC Genomics*). However, these conventional tools are not well adapted to high polyploids and thus many true SNPs at low frequency are not detected/or confused with sequencing errors. To optimize the SNP calling step to the polysomic high polyploidy context of sugarcane, we developed a method that is described in the Materials & Methods which allows us to distinguish SNPs at low

frequency from sequencing errors.

So far, researchers have been using sorghum as a reference to map sugarcane reads, but a large part of the reads did not map as expected because of the divergence between sorghum and sugarcane species and thus were lost for the analyses. So having a sugarcane reference genome will be a major improvement.

We believe that this sequence that represents, as most other plant reference sequences, one basic genome (although in this case it is a mosaic) will be very useful for many applications that require a framework for mapping reads of sequenced accessions: e.g. genetic analysis (QTL, GWAS,...) using genotyping by sequencing data, or genomic studies using WGS data. However, as always with sugarcane, these studies will require methods (e.g. SNP calling,) adapted to the high polyploid polysomic context of sugarcane.

As for LD and GWAS analysis, several teams including ours have developed and successfully implemented GWAS strategies for sugarcane (see for example: Jannoo et al 1999 Theor. Appl. Genet. 99 : 1053-1060 ; Raboin et al 2008 Theor. Appl. Genet 116(5):701-714 ; Wei et al. 2006 TAG and 2010 Genome ; Debibakas et al 2014 TAG). So far these strategies take account of presence/absence of SNP variants, disregarding dosage effects.

Another important application underway is to use our monoploid reference as a foundation for sequencing the whole genome of a cultivar. As all BACs were assembled in the absence of their homolog/homeologs and in small pools of 24 to 96 BACs, we avoided many of the problems of collapsing homolog/homeologs/paralogs and thus the high quality individual BAC sequence could be used as a control/reference in future whole genome assembly projects.

We modified the text to better explain this.

2. Following the previous comments, is it possible to map sequenced BACs to the genetic map generated in the paper, so these BACs can be separated to sub genomes?

As the two genomes can recombine and since in sugarcane all maps including this one are far from saturated, it would not be safe to attribute a species origin for the BAC based only on their position on the genetic map.

However, we exploited WGS data from *S. officinarum* vs *S spontaneum* to attribute an origin to the STP contigs using a method based on multiple allele origin discrimination to attribute an origin *S. officinarum* vs *S spontaneum* to the contigs of the STP: 1) for each locus, allele frequency comparison between the two reference germplasm samples taken to represent the two parental species, in order to assign a likely origin for individual SNP alleles, and; 2) analysis of likely-origin patterns among contiguous SNPs in order to take into account SNP allele chains (= haplotypes). This information has been added in the manuscript.

3. The placement of BACs basically relies on alignment of BAC end sequences. It can be confusing when placing BACs using sequence information only, since there are too many duplications and transposons in the genome. Genetics map is useful and complementary to the sequence-based tools. I would suggest the authors to take a look at the agreement between the genetic map and tilling path of 4535 BACs.

To select the MTP of BACs to be sequenced, BACs were located on the sorghum sequence based on an average of 10 WGP sequence tags per BAC (11,732 BAC anchored with 113,257 tags, including 90,953 tags that aligned at a single position on sorghum) and not based on

only two BAC-end sequences. In addition, after sequencing, all BAC sequences were re-mapped onto the sorghum genome based on their entire sequence length. We modified the text to better explain this.

4. In the smaller scale, I am wondering why the authors didn't barcode those STP BACs and sequence them independently. It would achieve higher assembly accuracy by doing so, just like those old-time but highly accurate reference genomes using the BAC-by-BAC approach.

For cost reasons, BACs were not bar-coded individually but the pools of BACs (of 24 or 96 BACs) were bar-coded. Also we carefully selected the BACs put into each pool to ensure that they did not overlap, thus avoiding assembly problems.

5. Two sequencing approaches were used to assemble BACs. If it is possible, I would compare the results of two approaches. For example, taking a few BACs which was sequenced by PacBio then sequence them using Illumina, then compare the results. It would be help to decide how to assemble a high-quality sugarcane genome in future, e.g. high density genetic markers + BACs sequences by Illumina or PacBio on the whole genome.

We did not conduct specific experiments to compare both methods, but when looking at the assembly (Supplementary Table 1) we observed that 85% of the BACs sequenced with PacBio were assembled in one contig, as opposed to 82% when sequenced with Illumina. In addition, we observed more BACs with 4 or more contigs with Illumina. PacBio was also chosen because long reads allow covering repeat sequence lengths and so lead to better assembly. In addition, the BACs were sequenced with high depth (an average of 75 x) to obtain high quality assemblies.

6. In line 281, it goes too far to announce that "is the first high quality assembly of sugarcane genome". In contrast, it needs to be emphasized the genome coverage is very limited and the monoplloid reference genome is a mixture of all sub genomes.

We modified this sentence.

The mosaic monoplloid genome encompasses BAC sequences corresponding to distinct homologous chromosomes/haplotypes from two subgenomes (*S. officinarum* and *S. spontaneum*), but the coverage of the basic genome is not very limited:

- i) the sequence covers most of the euchromatin/gene rich part of the basic genome
 - ii) it captures 79% (7,791) from a set of 9,871 conserved grass genes
- so a large part of the genes/locus are present in this reference and accurately captured.

In addition, the pangenome concept recently showed that not all genes are present in any accession in any crop, so any reference genome will miss dispensable genes.

Reviewer #2 (Remarks to the Author):

The authors have generated a reference resource of 4083 gene enriched BAC reference assemblies using PACBio sequencing for sugarcane. The strategy used 20,736 BAC end sequences, and aligned to sorghum to select a minimum tiling path 11,732 genes enriched BACs, plus 577 previously sequenced BACs. From these 4,660 were selected to represent a sequencing tiling path, of gene rich regions, representing one of the 8 homeologous regions. 4083 BACs were pooled, and then sequenced using PAC Bio, for a total of 531 MB of

assembled contigs. Contigs were mapped back to the individual BACs using initial tag sequence. Overlap between BACs were identified, trimmed and the contigs were then merged to a pseudomolecule based on sorghum order.

The authors present a large body of work, have been careful in their analysis, and have generated a valuable resource for the community.

As a genome resource I have concern regarding the representation of the mosaic nature of the pseudomolecules generated representing a heterogeneous mixture of Homeologous groups. This is very misleading giving the polyploidy nature of the genome. Would it be better to represent these as the individual BACs? And provide the tiling path information to be used versus pseudomolecules?

As a genome resource, we will provide access to the individual BAC sequences (on public databases and on the sugarcane genome hub that we created) and to the STP.

We designed the STP in order to use this reference as a framework for sequence read mapping. Because having overlaps between BAC/haplotypes would generate artificial duplication in the reference and would hamper proper read alignments.

We changed the term pseudomolecule by mosaic to super-scaffolds

See also the first comment to reviewer 1.

While the data is available from the French site the BAC contigs and annotations should be submitted to ENA or GenBank.

The BAC sequences and annotations are ready to be released through a public database as soon as the manuscript is accepted.

The authors discuss gene space and comparison to sorghum, but do not present statistics on the gene structure stats, length of transcript, exon, and intron, UTRs.

A Supplementary Table 2 has been added on gene structure statistics.

Is there any annotation on the TES and do they see differences between the two ancestral genomes identified by the mapping population.

The TEs have been annotated and their quantitative and qualitative distribution analyzed in the STP and in WGS data from the R570 cultivar, one *S. spontaneum* clone and one *S. officinarum* clone. A paragraph and a table have been added.

For the two ancestral genomes, based on the contigs available, can the authors say anything about the fractionation of genes between these?

Here we did not sequence series of overlapping BACs corresponding to distinct homologs or homeologs so we cannot look at the gene fractionation.

Moreover, sugarcane is a relatively “young” polyploid and several published studies have shown that gene content and order is very conserved between homologous and homeologous chromosome segments (Jannoo et al, 2007 in Plant J, Garsmeur et al, 2011 in New Phytol, Vilela et al, 2017 in Genome Biol Evol).

The author's state this will be a useful resource for the fine mapping, are they referring to the anchored BAC map or the sequencing itself? If the latter, it will be useful for only a small fraction of the genome, given the coverage of the orthologous group, while the BAC map can be addressed for fine mapping and walking if needed.

The reference sequence should facilitate the identification of candidate genes/loci in regions found to be associated with phenotypic traits by enabling us to locate these regions within the annotated genome. In addition, the reference sequence as a physical map could also be useful for developing fine mapping strategies.

The targeted allele or gene will not necessarily be present in the STP but this is true for any reference sequence and even more probable than previously anticipated with the recent concept of dispensable- core- and pan- genomes.

We changed this part in the manuscript to reflect this.

The authors also point to the use of analyses for expression data.

Were the authors able to provide insights on how the minimum genome space impacts transcription profiling studies, using the data sets generated on the project.

We did not perform specific analysis on this aspect but, as for other plants, the reference should be useful for transcriptomic studies by providing well-assembled gene structures covering ~80% of the genes in sugarcane. However as for other types of studies, this will require to adapt methods to the highly polyploid context of sugarcane

Major suggestions:

1) Submission of BACs & annotations to ENA.

The BAC sequences and annotations are ready to be released through a public database as soon as the manuscript is accepted

2) Gene statistics table comparison to sorghum. Consider using Sorghum V3 annotations recently published.

A table has been added as supplementary data

3) TE annotations and comparison to sub genomes

TE annotation and comparison were performed and are now reported in the manuscript

Minor suggestions:

1) Figure 1 was cited in the introduction, would recommend citing this as a result.

Figure 1 has been split into two new figures and repositioned as suggested

2) Review language Line 108-110

Genetic maps of this cultivar were constructed, and unpublished data, and aligned with sorghum.

This has been modified

3) Table 1 may be better represented as a follow, chart with detail numbers in a supplementary table.

The table has been simplified and part of the data put in the supplementary data

4) *Assembly summary and statistics could be moved to the assembly result section to give the readers a better overview.*

This has been modified

Reviewer #3 (Remarks to the Author):

The genome of sugarcane is difficult to assemble due to its polyploid, aneuploid and heterozygous features. To get around some of these difficulties, Garsmeur et al. have assembled a monoploid sugarcane genome using BAC sequencing and sorted the assembly into pseudomolecules based on synteny with sorghum. Construction of a genetic map using DArTseq markers showed the presence of chromosomal rearrangements. The manuscript is well-written and the level of English is high.

The methods used by the authors are sound and follow standard practice for genome assembly, though perhaps notably additional mapping technologies such as HiC and BioNano or a denser SNP-based genetic map may have allowed higher mapping resolution in complex areas.

Since the MTP contains BACs from distinct homologous and homeologous chromosomes, Hi-C or bionano would not bring useful information.

A denser SNP-map would have been useful but, given the genome structure of sugarcane (= polysomic high polyploid), only single dose markers can be used for mapping, which is why all sugarcane maps are not very dense so far. This is a major constraint in sugarcane. This map, however, will be the highest resolution sugarcane map published so far.

Sugarcane is an important crop and even though this is a monoploid assembly, it may be of interest to readers of Nature Communications, though currently it reads as a rather technical description of the assembly. The manuscript could be improved with additional biological analysis. For example, other recent genome assemblies published include analyses of intraspecific gene presence-absence variation, or additional analysis of key gene functions and traits, which in this case would include sugar production among others. By placing the genome assembly in a biological context while acknowledging the limitations of this not being a complete assembly would in my opinion improve the manuscript.

Sugar production would seem a logical candidate for functional analysis. However, sucrose content, like most major traits in sugarcane, is quantitatively inherited. This trait has been intensively studied through functional analyses and these studies have suggested that it is influenced by a very high number of genes (see for example the recent large transcriptomic study of Thirugnanasamban et al. 2018 in BMC Genomics).

So to add biological analyses, we analyzed sugarcane genes that were not found in sorghum. We also annotated TEs and analyzed their quantitative and qualitative distribution in the STP versus the whole genome of cultivar R570 and in *S. officinarum* versus *S. spontaneum*.

We also believe that the description of chromosome large structural variations that explain the distinct basic chromosome number in the two species involved in modern cultivars is new important information that brings new insight on the evolution of *Saccharum* genomes and their polyploidization timing.

Minor comments:

It will be good to specify the version of sorghum genome used.

This has been added

It will be good to specify the version of each software used, such as bowtie2, HGAP3, Quiver, BLAST, BLAT and Trinity.

This has been added

In bowtie2 parameter settings on line 312, the '-i' option should be S, 1, 0.75 instead of S1, 0.75?

This has been modified

L45 Reference recommended.

A reference has been added

L59 Parentheses around "review in 11". Should read "reviewed in"?

This has been modified

L66 "resistant" should be "resistance"?

This has been modified

L313-318 Can be confusing to read, should be reworded for clarity.

This has been modified

Reviewers can access the sequence and annotation data through the sugarcane genome hub:

<http://sugarcane-genome.cirad.fr/>

user : reviewer

password: 07fI510J

This hub will be improved over time as we are doing for our banana genome hub (<http://banana-genome-hub.southgreen.fr/>)

REVIEWERS' COMMENTS:

Reviewer #1 (Remarks to the Author):

Comments for the manuscript entitled "A monoploid reference sequence for the highly complex genome of sugarcane"

The sugarcane genome certainly lies at the extreme of complexity spectrum across all crops. In the earlier version of this manuscript, I raised concerns about the usefulness about the monoploid reference genome. The authors took my concerns into account and address them well. The manuscript was revised properly. The analysis about gene content and transposon was added as well, giving a more complete picture about the genome. The authors have greatly improved the manuscript. I have just one suggestion:

1. The polysomic nature of sugarcane challenges high-quality assembly of sugarcane genome. The monoploid reference genome is a good start. However, the monoploid assembly in the manuscript is quite different from classical ones in diploid species. To avoid being misleading, would it be better to change the title as "A mosaic monoploid reference sequence for the highly complex genome of sugarcane"?

Reviewer #2 (Remarks to the Author):

The authors have responded to all of the requests for modifications with the exception of the following exception. This is a testable hypothesis and if the recommendation is this provides improvements, over transcripts assemblies, it would have been good to demonstrate this with existing data sources as this is being submitted as a community resource, as it is unclear if this is a more robust path than using the transcript assemblies.

"The authors also point to the use of analyses for expression data. Were the authors able to provide insights on how the minimum genome space impacts transcription profiling studies, using the data sets generated on the project."

Reviewer #3 (Remarks to the Author):

We thanks the reviewers for producing an improved manuscript which fully addresses each of our previous comments. There is only one minor question regarding editorial policy. It is stated in the editorial policy checklist that custom software or code was used. If this is the case, a code availability statement should be included in the manuscript. I see a data availability statement but no code availability statement. Could the authors clarify if custom code was used, and if so, please include a code availability statement.

RESPONSE TO REVIEWERS' COMMENTS

Reviewer #1 (Remarks to the Author):

Comments for the manuscript entitled “A monoploid reference sequence for the highly complex genome of sugarcane”

The sugarcane genome certainly lies at the extreme of complexity spectrum across all crops. In the earlier version of this manuscript, I raised concerns about the usefulness about the monoploid reference genome. The authors took my concerns into account and address them well. The manuscript was revised properly. The analysis about gene content and transposon was added as well, giving a more complete picture about the genome. The authors have greatly improved the manuscript. I have just one suggestion:

1. The polysomic nature of sugarcane challenges high-quality assembly of sugarcane genome. The monoploid reference genome is a good start. However, the monoploid assembly in the manuscript is quite different from classical ones in diploid species. To avoid being misleading, would it be better to change the title as “A mosaic monoploid reference sequence for the highly complex genome of sugarcane”?

Response:

The title of our manuscript has been modified as suggested. “A mosaic monoploid reference sequence for the highly complex genome of sugarcane”

Reviewer #2 (Remarks to the Author):

The authors have responded to all of the requests for modifications with the exception of the following exception. This is a testable hypothesis and if the recommendation is this provides improvements, over transcripts assemblies, it would have been good to demonstrate this with existing data sources as this is being submitted as a community resource, as it is unclear if this is a more robust path than using the transcript assemblies.

“The authors also point to the use of analyses for expression data.

Were the authors able to provide insights on how the minimum genome space impacts transcription profiling studies, using the data sets generated on the project.”

Response:

In the second version of the manuscript we submitted, we removed the comment on functional/ expression study. So, we do not anymore “point to the use of analyses for expression data »

We only wrote “This monoploid reference sequence provides an essential genome template for aligning sequencing data, such as genotyping by sequencing, WGS and RNAseq data to

explore hom(oe)ologous allelic variation and perform genetic (e.g. QTL, GWAS) and genomic studies in cultivars and sugarcane germplasm”

We removed this reference to expression/functional studies because although, we believe that having access to one well assembled allelic version for the genes present in the monoploid sequence would be useful, in particular because building a reference transcriptome of quality through de-novo assembly is challenging in polyploids (Bin He et al in BMC Genomics, 2015), we presently do not have good data to well demonstrate it.

He B, Zhao S, Chen Y, Cao Q, Wei C, Cheng X, Zhang Y (2015) Optimal assembly strategies of transcriptome related to ploidies of eukaryotic organisms. BMC Genomics 16:65

Reviewer #3 (Remarks to the Author):

We thanks the reviewers for producing an improved manuscript which fully addresses each of our previous comments. There is only one minor question regarding editorial policy. It is stated in the editorial policy checklist that custom software or code was used. If this is the case, a code availability statement should be included in the manuscript. I see a data availability statement but no code availability statement. Could the authors clarify if custom code was used, and if so, please include a code availability statement.

The custom pipeline that was developed to perform SNP analyses has been deposited on GitHub web-site at <https://github.com/SouthGreenPlatform/VcfHunter/>. A documentation explaining how to use the tools is also available.

We have included in the ‘Methods’ section a code availability statement.